# Sync4D: Video Guided Controllable Dynamics for Physics-Based 4D Generation

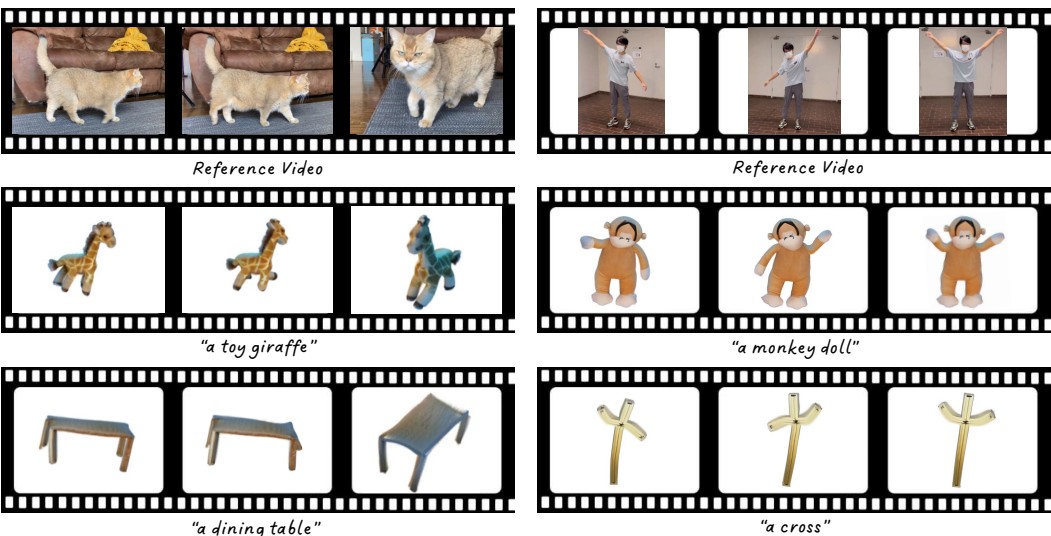

Figure 1: Our proposed method can create dynamics on various generated 3D Gaussians guided by the reference casual video.

## ABSTRACT

In this work, we introduce a novel approach for creating controllable dynamics in 3D-generated Gaussians using casually captured reference videos. Our method transfers the motion of objects from reference videos to a variety of generated 3D Gaussians across different categories, ensuring precise and customizable motion transfer. We achieve this by employing blend skinning-based non-parametric shape reconstruction to extract the shape and motion of reference objects. This process involves segmenting the reference objects into motion-related parts based on skinning weights and establishing shape correspondences with generated target shapes. To address shape and temporal inconsistencies prevalent in existing methods, we integrate physical simulation, driving the target shapes with matched motion. This integration is optimized through a displacement loss to ensure reliable and genuine dynamics. Our approach supports diverse reference inputs, including humans, quadrupeds, and articulated objects, and can generate dynamics of arbitrary length, providing enhanced fidelity and applicability. Unlike methods heavily reliant on diffusion video generation models, our technique offers specific and high-quality motion transfer, maintaining both shape integrity and temporal consistency.

## 1 INTRODUCTION

The introduction of large-scale diffusion-based generative models (Rombach et al., 2022; Saharia et al., 2022) has sparked a revolution in creative and high-quality image synthesis, which has been successfully extended to video generation (Blattmann et al., 2023; Chen et al., 2024; Xing et al.,

2023) and further evolved into 3D generation (Poole et al., 2022; Lin et al., 2023; Chen et al., 2023; Wang et al., 2024; Shi et al., 2023; Li et al., 2024a; Liang et al., 2024; Liu et al., 2023; Raj et al., 2023; Tang et al., 2024), laying the groundwork for dynamic 3D content or 4D generation. This technological convergence enhances various applications, from virtual reality to simulation training, by significantly boosting the realism and interactivity of virtual environments.

However, despite these technological strides, existing methodologies still face significant limitations. Current implementations, utilizing Score Distillation Sampling (SDS) (Poole et al., 2022) as seen in (Bahmani et al., 2024b; Ling et al., 2024; Singer et al., 2023; Zheng et al., 2024; Bahmani et al., 2024a), aim to distill motion priors from video diffusion models to facilitate dynamic 3D creation. However, this often leads to inaccurate motion representations. Alternatively, methods like those documented in (Yin et al., 2023; Ren et al., 2023) directly use the per-frame outputs from video diffusion models as references. While faster and more straightforward, this approach still fails to adequately address issues of movement irrationality and shape incoherence in the generated outputs. The effectiveness of both approaches is inherently limited by the capabilities of the pretrained video diffusion models they adopted. Therefore, the generation quality of the dynamic and geometry quality frequently suffers from inconsistencies and poor geometric integrity. Moreover, these methods lack precise motion control, typically relying on vague text prompts to guide motions, which further compromises the fidelity and applicability of the generated content.

Significant advancements have also been made in dynamics representation, particularly in integrating physical properties into dynamic models. The introduction of PhysGaussian (Xie et al., 2024), which utilizes a novel style of 3D Gaussians representation from Kerbl et al.(Kerbl et al., 2023), has facilitated high-quality motion synthesis. Zhang et al.(Zhang et al., 2024) pioneered the integration of dynamic generation model with physical simulation techniques (Hu et al., 2018a; Xie et al., 2024), marking a crucial step forward in this domain. Incorporating physical simulation produces more reliable and genuine dynamics on 3D Gaussian representations. However, these methods require hand-crafted input motions, which are also limited to a narrow range of actions and relatively simple scenarios.

In this work, we introduce a novel approach for creating controllable dynamics in generated 3D Gaussians guided by casually captured reference videos. As shown in Figure 1, our method transfers the motion of an object from the reference video to various generated 3D Gaussians across different categories. To achieve this, we first apply blend skinning-based non-parametric shape reconstruction to extract the shape and motion of the reference object from the video. This process allows the decomposition of the reference object into motion-related parts based on skinning weights. Next, we establish shape correspondences between the reference shape and the generated target shapes utilizing pretrained 2D diffusion models and 3D point cloud models. Finally, we map the motion-related parts to the corresponding target shapes, enabling the matched parts in the target shapes to inherit the motion from the reference object parts.

To tackle the shape and temporal inconsistency issue that widely appears in existing works, instead of the commonly used point-wise deformation, we drive the target shapes with the matched motion using Material Point Method (MPM) physical simulation (Hu et al., 2018a; Xie et al., 2024; Zhang et al., 2024). However, due to the shape variation in target objects, directly providing the reference motion as input on each part to the physical simulation model may not produce the desired outputs and may suffer from cumulative errors. Therefore, we model a delta velocity field to adjust the input motion adopted from the reference, which is optimized by a displacement loss between two object spaces.

In summary, our contributions are as follows:

- We introduce a novel method that transfers motion from casually captured videos to various 3D-generated Gaussians, ensuring precise and customizable dynamics across different categories.

- Our technique employs shape reconstruction to extract shape and motion from reference objects. We segment the reference objects into motion-related parts based on skinning weights and map the parts to generated target shapes by establishing shape correspondences.

- We integrate physical simulation to drive target shapes with matched motion to ensure shape integrity and temporal consistency. Our approach further ensures reliable and genuine dy-

namics by introducing a displacement loss to optimize physical signals, avoiding cumulative errors.

- Our method supports diverse reference inputs, including humans, quadrupeds, and articulated objects. Unlike existing methods reliant on diffusion video generation models, our approach generates dynamics specific to the reference input and can be of arbitrary length.

## 2 RELATED WORKS

### 2.1 4D GENERATION

Dynamic generation seeks to create robust and persistent 3D representations that excel in virtual environments like gaming, animation, and virtual reality. Initiatives commonly begin with a text prompt specifying the 3D object and its motions (Bahmani et al., 2024b; Singer et al., 2023; Zheng et al., 2024). Zhao et al. (Zhao et al., 2023) adopt a different strategy, using an image prompt, which offers greater versatility over the 3D object's representation. Meanwhile, Yin et al. (Yin et al., 2023) and Ren et al. (Ren et al., 2023) utilize videos generated from video diffusion models as direct references, indicating that controlling motions through video input holds promise. However, these approaches face challenges, including constrained motion expression, discrepancies between the input text and the resulting motions, and poor generation results.

### 2.2 SHAPE AND MOTION RECONSTRUCTION FROM VIDEOS

Dynamics reconstruction from video footage is a prolonged and challenging endeavor, and reconstructing from monocular video poses an even greater difficulty. A commonly employed approach (Attal et al., 2023; Kratimenos et al., 2023; Pumarola et al., 2021; Li et al., 2023; Park et al., 2021a;b; Liu et al., 2022; Wang et al., 2023) involves utilizing a deformation field (Pumarola et al., 2021) to enhance the neural radiance field (Mildenhall et al., 2021) while concurrently implementing various techniques to ensure high-quality reconstruction. While these works mostly rely on multi-view datasets, Yang et al.(Yang et al., 2022; 2023c; Song et al., 2023c; Yang et al., 2023a) focus on reconstructing shapes from casual videos, achieving remarkable progress in the area. As 3D Gaussian Splatting proved to be an efficient and effective approach for reconstructing tasks, several works (Li et al., 2024b; Yu et al., 2024; Lin et al., 2024; Wu et al., 2024; Yang et al., 2024; Luiten et al., 2023; Lu et al., 2024) are adapted to dynamics reconstruction, achieving promising results.

### 2.3 MOTION TRANSFER

A common perspective on attaining reliable motion is to derive it from a real video and transfer it to another object. This can be achieved by estimating poses frame-by-frame and subsequently transferring these poses. However, these works (Doersch & Zisserman, 2019; Song et al., 2021; Chen et al., 2022; Song et al., 2023b) fundamentally rely on correspondences between the same category of objects. An alternative approach (Yatim et al., 2024; Park et al., 2024) to motion transfer based on the diffusion model has garnered popularity in the video domain. These methods can transfer motions between different types of objects. However, the quality of the results significantly falls short of the requirements for 3D and 4D generation, considering the inconsistency and vagueness of the video.

## 3 METHOD

We propose a framework capable of transferring motion from casually captured videos to generated static 3D objects, as illustrated in Figure 2. We begin by reconstructing the shape of the captured object from a video and extracting the motion information. In the subsequent stage, the reconstructed object will be matched with the target 3D Gaussian representation to achieve regional correspondence. Finally, we transfer the original motion to the corresponding target regions and utilize physics simulation to animate the 3D object. We optimize the velocity field in physics simulation by minimizing spatial displacement differences to enhance motion correctness, thereby achieving superior visual fidelity.

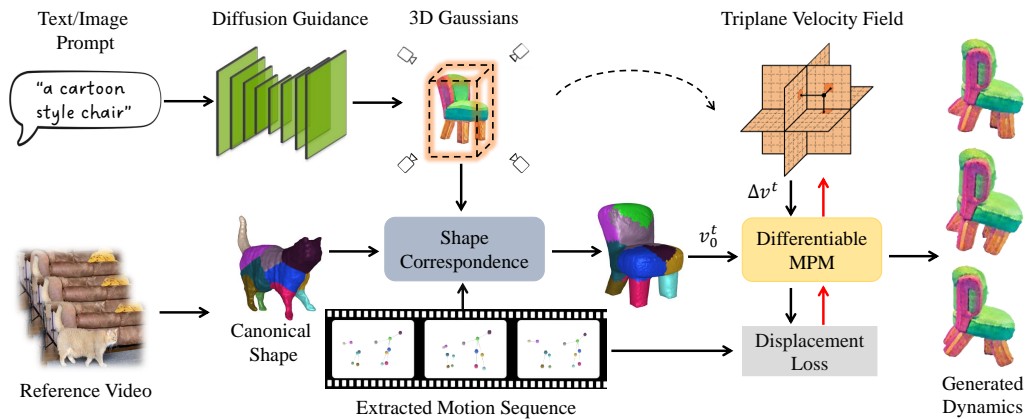

Figure 2: **Overview of Sync4D:** Sync4D processes a reference video to derive a canonical shape and a bone-based motion sequence through reconstruction techniques. Meanwhile, given a text prompt or image prompt, we generate a 3D Gaussian object through diffusion models. The framework matches motion-related parts from the reconstructed shape to the generated shape and transfers the motion. This motion information is then initialized into the velocity physical signals. We employ a triplane representation to produce a delta velocity field to adjust physical signals. The velocity field for each part of the target is optimized using the differentiable Material Point Method (MPM) simulation. To ensure fidelity to the original, a displacement loss is designed to reduce cumulative errors and ensure plausible motions.

## 3.1 PRELIMINARIES

Material Point Method (MPM) is a computational technique for simulating the behavior of continua. It uses a dual representation where material properties and state variables are stored on particles while computations and interactions are handled on a background computational grid. Following Phys-Gaussian (Xie et al., 2024), we employ MPM simulation directly on Gaussian particles, discretizing the entire scene into a set of Lagrangian particles. At timestep $t$, each particle $p$ maintains its state variables, which include spatial position $\boldsymbol{x}_p^t$, velocity $\boldsymbol{v}_p^t$ and its material properties, including mass $\boldsymbol{m}_p^t$, deformation gradient $\boldsymbol{F}_p^t$, Kirchhoff stress $\boldsymbol{\tau}_p^t$, affine momentum $\boldsymbol{C}_p^t$.

MPM simulation process transfers data between particles and grid nodes at each simulation period $\boldsymbol{\Delta t}$, which can be delineated into three distinct steps. Firstly, we apply particle-to-grid to transfer momentum as follows:

$$\boldsymbol{m}_i^t = \sum_p N(\boldsymbol{x}_i - \boldsymbol{x}_p^t)\boldsymbol{m}_p, \tag{1}$$

$$\boldsymbol{m}_i^t \boldsymbol{v}_i^t = \sum_p N(\boldsymbol{x}_i - \boldsymbol{x}_p^t)\boldsymbol{m}_p(\boldsymbol{v}_p^t + \boldsymbol{C}_p^t(\boldsymbol{x}_i - \boldsymbol{x}_p^t)). \tag{2}$$

Here $\sum_p N(\boldsymbol{x}_i - \boldsymbol{x}_p^t)$ is the B-spline kernel, and $\boldsymbol{v}_i^t$ is the updated velocity on grid node. Then we use grid transfer to get the next state grid velocity $\boldsymbol{v}_i^{t+1}$ as

$$\boldsymbol{v}_i^{t+1} = \boldsymbol{v}_i^t - \frac{\boldsymbol{\Delta t}}{\boldsymbol{m_i}}(\sum_{\boldsymbol{p}} N(\boldsymbol{x}_i - \boldsymbol{x}_p^t)\frac{4}{r^2}V_p^0 \frac{\partial \psi}{\partial \boldsymbol{F}} \boldsymbol{F_p^t}(\boldsymbol{x}_i - \boldsymbol{x}_p^t) + g_i^t), \tag{3}$$

where $r$ is the grid resolution, $V_p^0$ is the initial representing volume, $\psi$ is a strain energy density function related to Kirchhoff stress $\boldsymbol{\tau}_p^t$, $g_i^t$ is a possible external force. Finally, we convert the grid velocity to particle velocity at timestep $t + 1$, alongside transferring of particle positions:

$$\boldsymbol{v}_p^{t+1} = \sum_i N(\boldsymbol{x}_i - \boldsymbol{x}_p^t)\boldsymbol{v}_i^{t+1}, \quad \boldsymbol{x}_p^{t+1} = \boldsymbol{x}_p^t + \boldsymbol{\Delta t}\boldsymbol{v}_p^{t+1}. \tag{4}$$

Since our work mainly focus on optimizing velocity field $\boldsymbol{v}(p,t)$, material properties $\boldsymbol{F}_p^t$, $\boldsymbol{\tau}_p^t$, and $\boldsymbol{C}_p^t$ update are not listed here. Please refer to Appendix A.1 for more information on the MPM simulation process.

## 3.2 EXTRACTING SHAPE AND MOTION FROM VIDEOS

To extract the shapes and motions of arbitrary objects from casual videos, we model the object with bones and neural blend skinning (Jacobson et al., 2014) following several existing non-parametric reconstruction methods (Yang et al., 2022; Song et al., 2023a; Yang et al., 2023b;c; Song et al., 2024). For a point $\mathbf{x}^t$ in three-dimensional space at time $t$, we aim to determine its equivalent point $\mathbf{x}^*$ within a canonical space. The model achieves the transition between $\mathbf{x}^t$ and $\mathbf{x}^*$ by incorporating the rigid transformations linked to the coordinates of bones in 3D. We define $\mathbf{G}^t \in SE(3)$ as the global transformation mapping the entire structure from the fixed frame to time $t$. We initialize the canonical bone center coordinates $\mathbf{B}^* \in \mathbb{R}^{B \times 3}$ and let $\mathbf{J}_b^t \in SE(3)$ indicate the relative rigid transformation adapting the $b$-th bone from its initial position $\mathbf{B}_b^*$ to its transformed state $\mathbf{B}_b^t$ at time $t$. These transformations can be described by the following relations:

$$\mathbf{x}^t = \mathcal{W}^{t,\rightarrow}(\mathbf{x}^*) = \mathbf{G}^t\mathbf{J}^{t,\rightarrow}\mathbf{x}^*, \tag{5}$$

$$\mathbf{x}^* = \mathcal{W}^{t,\leftarrow}(\mathbf{x}^t) = \mathbf{J}^{t,\leftarrow}(\mathbf{G}^t)^{-1}\mathbf{x}^t, \tag{6}$$

where $\mathcal{W}^{t,\rightarrow}$ and $\mathcal{W}^{t,\leftarrow}$ indicate forward and backward warping, $\mathbf{J}^{t,\rightarrow}$ and $\mathbf{J}^{t,\leftarrow}$ represent the weighted averages of $B$ rigid transformations $\{\mathbf{J}_b^t\}_{b \in \{1,...,B\}}$, mapping the bones from their default positions to their current configurations at time $t$. Since the primary aim of the reconstruction is to offer motion cues for the target objects, we configure the number of bones $B$, to be the minimum count of articulated segments required to accurately model the reference shape.

The skinning weights are defined as $\mathbf{W} = \{w_1, ..., w_B\} \in \mathbb{R}^B$. For any 3D point $\mathbf{x}$, the skinning weights are calculated using the Mahalanobis distance $d_M(\mathbf{x}, \mathbf{B}^t)$ between the point and the Gaussian-shaped bones under pose $\mathbf{B}^t$, as indicated in the equation:

$$\mathbf{W} = \text{softmax}(d_M(\mathbf{x}, \mathbf{B}^t) + \mathbf{W}_\Delta). \tag{7}$$

where $\mathbf{W}_\Delta$ is produced by a coordinate MLP to enhance the details. We optimize all the parameters following the framework of BANMo (Yang et al., 2022).

## 3.3 PART MAPPING WITH SHAPE CORRESPONDENCE

To transfer the motion, we map the articulated parts from the reference shape to the target shape. We first extract the surface meshes of the shapes. We abuse the notation to define the vertices of the reference mesh and target mesh as $\mathbf{X}^{ref} \in \mathbb{R}^{N_{ref} \times 3}$ and $\mathbf{X}^{tar} \in \mathbb{R}^{N_{tar} \times 3}$. Inspired by Diff3F (Dutt et al., 2023), we utilize pretrained 2D diffusion models to obtain the 2D semantic features on multi-view renderings and back-project to 3D vertices to get $f_{diff} \in \mathbb{R}^{N \times 1024}$. However, solely using semantic features may not provide enough information, for example, it cannot distinguish the different limbs of humans and quadrupeds. Therefore, we adopt another geometry based pretrained 3D correspondence network (Zeng et al., 2021) to extract additional features $f_{geo} \in \mathbb{R}^{N \times 128}$, the resulting features on mesh surfaces are given by:

$$f^{ref} = f_{diff}^{ref} \| f_{geo}^{ref}, \quad f^{tar} = f_{diff}^{tar} \| f_{geo}^{tar} \tag{8}$$

Where $\|$ denotes concatenation. We segment the reference objects into $B$ articulated parts based on the optimized skinning weights. The part labels are noted as $\mathbf{Y}^{ref} \in \mathbb{R}^{N_{ref}}$, the label for vertex $n$ is obtained:

$$y_n^{ref} = \arg\max(\mathbf{W}(\mathbf{X}_n)) \tag{9}$$

Then, we calculated the mean feature for each part of the reference object:

$$\bar{f}_b^{ref} = \frac{1}{N_b} \sum_{n:y_n^{ref}=b} f_n^{ref} \tag{10}$$

We derive the correspondence between each vertex in the target mesh and the reference part as:

$$y_n^{tar} = \arg\max_{b \in B}\left(\frac{\bar{f}_b^{ref} \cdot f_n^{tar}}{\|\bar{f}_b^{ref}\|\|f_n^{tar}\|}\right) \tag{11}$$

We further perform an outlier removal based on the distance to part centroids to get $\hat{y}_n^{tar}$. From the mapped surface points $\hat{y}_n^{tar}$, we can draw bounding boxes for each part and assign all the Gaussian points in the bounding boxes to the corresponding part. The relative motion for $b$-th part can be approximated as $\Delta \mathbf{B}_b^t = \mathbf{B}_b^{t+1} - \mathbf{B}_b^t$.

### 3.4 Physics-Integrated Motion Transfer

The process of motion transfer commences with the utilization of the reconstructed prior alongside the identified corresponding matching. This is achieved through the initialization of $\boldsymbol{v}$ at the onset of each simulation, guided by the motion sequence observed in reference space, broadly indicating the velocity direction. The initialized velocity for $b$-th part of target should be:

$$v_0^t = \hat{v^t} = \frac{\hat{\delta}^t}{\boldsymbol{N \Delta t}}, \;\; \hat{\delta}^t = \mathbf{b}^{t+1} - \mathbf{b}^t,$$ (12)

where $\mathbf{b}$ represents $\mathbf{B}_b$. In this section, we drop $b$ in every notation for simplicity.

To better control the simulated motion and avoid cumulative errors, we employ a triplane representation (Chan et al., 2022) accompanied by a three-layer MLP to adjust the velocity field. The network shares the same spatial information as the physics field, generating particle-level $\Delta v$ for each part of the object. The velocity field before simulation can then be set to:

$$\boldsymbol{v}^t \leftarrow v_0^t + \Delta v^t.$$ (13)

Based on the given velocity states and other physics properties, we animate the 3D static generation with a differentiable MLS-MPM (Hu et al., 2018a) simulator. This process should be done between adjacent two frames, estimating one motion sequence, which can be formulated as follows:

$$\boldsymbol{x^{t+1}, v^{t+1} = S(x^t, v^t, \theta, \Delta t, N)},$$ (14)

where $\boldsymbol{x}^t$ denotes particle positions of $b$-th part at time $t$, and similarly $\boldsymbol{v}^t$ denotes the velocities of corresponding particles at time $t$. $\boldsymbol{\theta}$ denotes the collection of the physical properties of all particles: deformation gradient $\boldsymbol{F}^t$, gradient of local velocity fields $\boldsymbol{C}^t$, mass $\boldsymbol{m}$, Young's modulus $\boldsymbol{E}$, Poisson's ratio $\boldsymbol{\nu}$, and volume $\boldsymbol{V}$. $\Delta t$ is the simulation step size, and $N$ is the number of steps.

While the modification goal is to ensure that the resulting pose closely matches the reconstructed one, one approach to addressing this issue is to approximate the displacement in the target space to be consistent with the displacement in the reference space, considering the respective part sizes. With this as a reference, we optimize velocity field $\boldsymbol{v}$ for all parts by a per-frame loss function:

$$L_x^t = \sum_b L_1(\delta_b^t - \frac{s_t}{s_o}\hat{\delta}_b^t),$$ (15)

where $s_t, s_o$ is the coverage ratio for target space and reference space, respectively. To calculate the displacement $\delta$, we determine the positional difference between the part mass centroid of the initial state and the simulated end state, which is slightly divergent from the initialization of velocity.

Furthermore, we employ total variation regularization across all spatial planes to promote spatial continuity. Denoting $u$ as one of the 2D spatial planes and $u_{j,k}$ as a feature vector on the 2D plane, the total variation regularization term is formulated as:

$$L_{tv}^t = \sum_{j,k} \|u_{j+1,k} - u_{j,k}\|_2^2 + \|u_{j,k+1} - u_{j,k}\|_2^2$$ (16)

Rather than directly training the complete video motion, we utilize the motion between two frames as the training phase. Subsequently, after sufficient training in this phase, we advance to the next motion phase. This training methodology ensures that the dynamics' posture is as accurate as possible after each motion sequence. After training the relative motion, we apply the global transformation $\mathbf{G}^t$ on the entire 3D Gaussians for each frame to get the final rendering.

## 4 Experiments

In this section, we demonstrate the versatility of our framework for generalized data and substantiate the reliability of the resulting motions.

## 4.1 Experimental Settings

**Implementation details**. For text-to-3D generation, we choose LucidDreamer (Liang et al., 2024) as our model, while for image-to-3D generation, we choose LGM (Tang et al., 2024) as our model. Our reconstruction model is implemented based on Lab4D (Yang et al., 2022; 2023a). We set the number of bones $B = 11$ for human, $B = 13$ for quadrupeds and $B = 2$ for laptops. For humans and quadrupeds, we provide an average initial bone center coordinates for faster training. For laptops, the bones are all initialized from the origin. The Gaussian objects from two generative models are viewed as our simulation area, which has 1.5 to 2 million particles for LucidDreamer generation and 20 to 50 thousand particles for LGM. Considering simulation consumption, we use a $41^3$ resolution grid to downsample LucidDreamer output, ensuring consistency with the LGM output by order of magnitude. We take the average coordinate of all particles within the same grid as our control point, where physical simulations are applied. Upon completion of the simulation, particles within the same grid point will share the same velocity field properties, ensuring the rigid body motion of the object.

For the optimization process, we utilize a triplane (Chan et al., 2022; Peng et al., 2020) followed by a three-layer MLP, similar to PhysDreamer (Zhang et al., 2024). Although we did not optimize the material properties, in our experiments, they retain physical significance and are adjustable. Users can select Young's modulus $E$ between $1 \times 10^3$ and $1 \times 10^5$, and the Poisson's ratio $\nu$ between 0.1 and 0.5, based on the desired visual effects. A higher $E$ results in a more resilient object, while a higher $\nu$ leads to a stiffer object.

We train our task on a single NVIDIA RTX 6000 Ada machine. Our training process requires 7-8 NVIDIA RTX 6000 Ada GPU minutes per frame, with an approximate memory consumption of 24 GB.

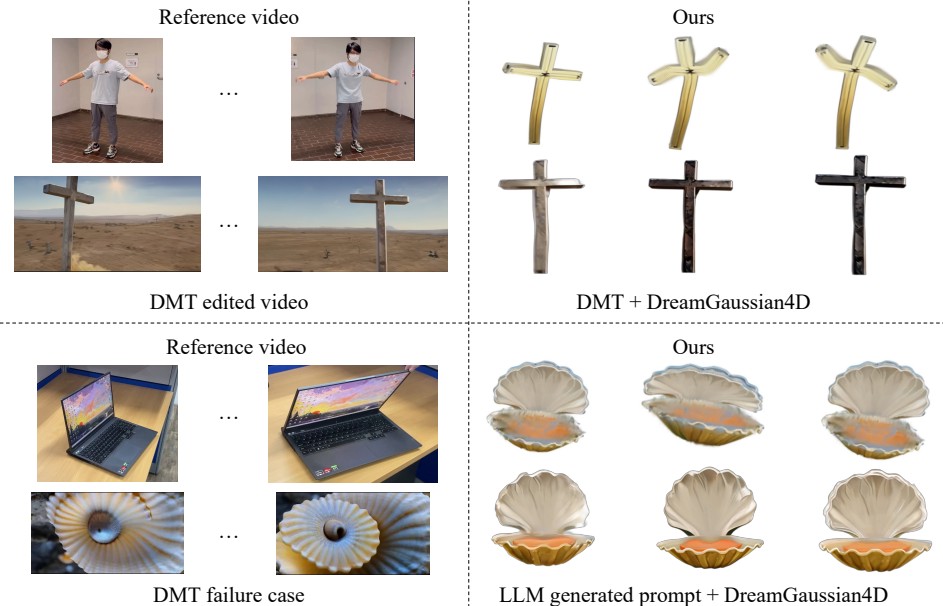

Figure 3: Comparative Analysis between Sync4D and Other Frameworks. On the left, the reference video alongside the edited video from DMT is displayed. The upper example shows a successful adaptation, whereas the lower example is deemed a failure due to continual alterations in shape and appearance across frames. On the right, the Sync4D outputs are highlighted, showcasing superior motion and shape consistency relative to other frameworks.

**Metrics**. Our framework focuses on the realism and similarity between input video motion and generated motion. For evaluation, we conduct a user study listing our results and the other experimental results as a pair. Three questions are set for better evaluation: the overall generation quality of the dynamic scene, the motion similarity of the input video and the 4D generation, and the shape consistency of results. We conduct the evaluation on three pairs and recruit 34 participants to join the

evaluation, getting a high score for all of the questions. Detailed experimental results can be referred at Appendix A.2

## 4.2 RESULTS

**Comparison with Generation Pipeline.** We compare our proposed method with one generation framework: video motion transfer (DMT) (Yatim et al., 2024) combined with DreamGaussian4D (Ren et al., 2023). The compared approach involves generating a motion-transferred video from the input casual video. This process begins by applying the DMT model to the initial video, effectively transferring the motion patterns to a new text-prompt object. Subsequently, the motion-transferred video is utilized in the DreamGaussian4D framework to generate the corresponding dynamics.

However, we observe in some complicated cases, the edited video from the DMT model has low quality and inconsistency. To tackle this problem, we employ ChatGPT (OpenAI, 2024) to extract the description of the original video and convert the subject term to our target object. Then, we input the description to DreamGaussian4D to obtain corresponding dynamics.

As Figure 3 illustrated, for both experiments, our results outperform in both motion similarity and shape consistency.

**Comparison with Pose Transfer Pipeline.**
Most 3D object animation techniques rely on skeletal structures. State-of-the-art automatic rigging and skeleton generation methods are predominantly trained on existing 3D assets, such as humanoid characters and animals. However, with the advent of 3D generation techniques capable of producing out-of-domain, creative assets, these methods often struggle to generalize effectively. For instance, as demonstrated in our tests (see Appendix Figure 9), auto-rigging methods like RigNet(Xu et al., 2020) perform poorly on non-standard objects, particularly those outside their training domain, such as creatively shaped assets generated by 3D algorithms.

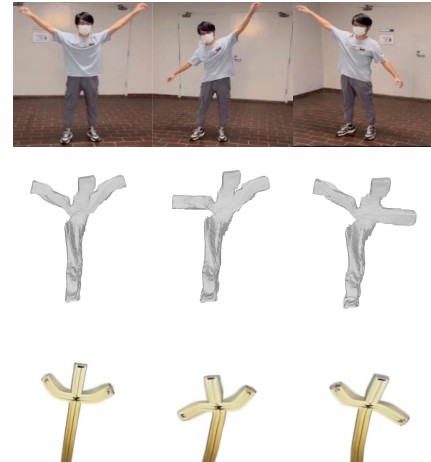

We also investigated commercial auto-rigging tools, including Mixamo(Adobe, 2024) and Anything World(AnythingWorld, 2024). Mixamo is limited to humanoid models and requires manual joint annotation, while Anything World only supports a narrow range of categories, such as humanoids, quadrupeds, and insects. Both tools

Figure 4: Comparison between novel pose transfer method (middle) and ours (bottom).

demand high-quality meshes and often fail to handle AI-generated 3D shapes, even after remeshing.

Additionally, we compared our proposed method with the skeleton-free pose transfer technique by (Liao et al., 2022), which, like others, is trained on conventional 3D assets and struggles with non-character objects. Notably, our approach successfully transfers human motion to non-standard objects, such as a Christian cross, demonstrating versatility beyond humanoid figures. Detailed comparative results are provided in Figure 4, illustrating the robustness of our method across diverse scenarios.

**Matching Results.**

Moreover, our matching method can handle correspondences between objects with different poses, fully demonstrating the robustness of our approach. Additionally, we present an example of a matching failure case, which leads to incorrect dynamic results.

All the matching details can be found in Appendix A.3.

**Overall Results.** We also present the qualitative results of our generated 3D dynamics in comparison with reference video frames In Figure 5. Our method effectively captures the reference motion while

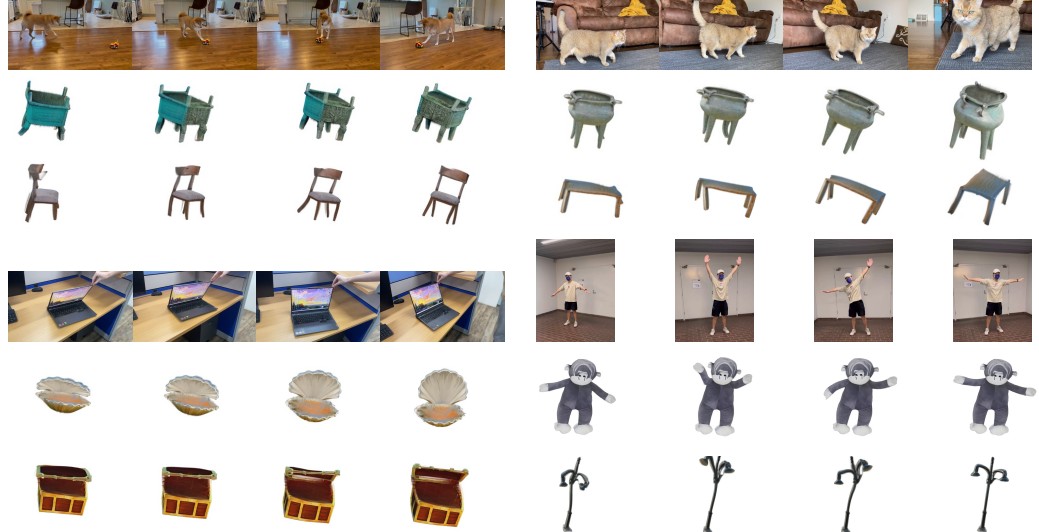

Figure 5: We present the qualitative results of our generated 3D dynamics with reference video frames. Our method generates dynamics that align with the reference motion while retaining the shape integrity and temporal consistency. Please check the video results in the supplementary materials for a more intuitive illustration.

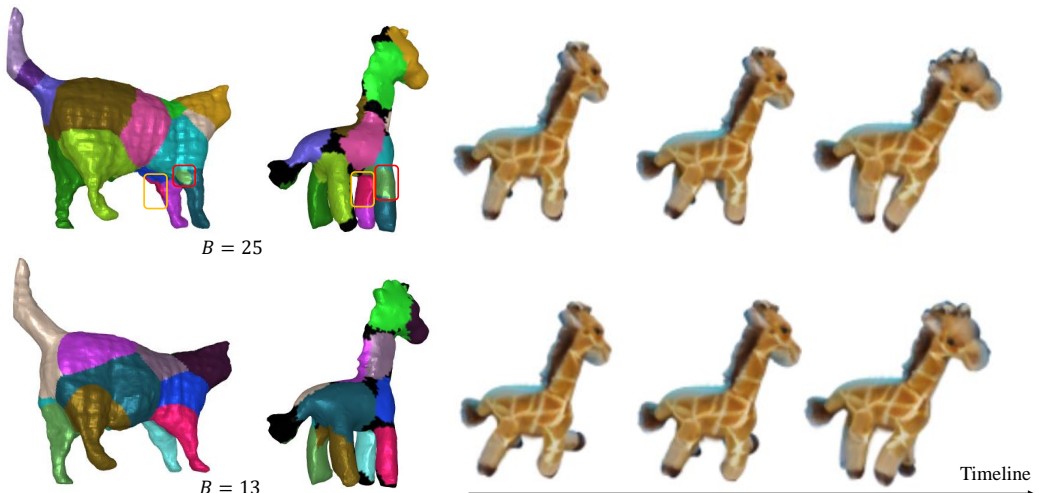

Figure 6: Ablation study on the number of bones in reconstruction to segment motion-related parts. **Upper Row:** number of bones $B = 25$. **Bottom Row:** number of bones $B = 13$, indicating the minimum articulated parts. Color black indicates removed outliers.

preserving both the integrity of the shape and the temporal consistency of the dynamics. Please refer to Appendix A.4 for more scenarios and the supplementary materials for video results.

### 4.3 ABLATION STUDIES

**Number of Motion-related Parts.** As illustrated in Figure 6, the upper row presents the matching and simulation results with the number of bones $B = 23$, close to the conventional settings in the SMPL (Loper et al., 2023) and SMAL (Zuffi et al., 2017). We observe that some parts might be redundant in modeling the motions, for example, the circled part near the creaking nest, which results

in stiffness in the target motion. In the bottom row, we set the number of bones to $B = 13$, indicating the minimum articulated parts, which produces better dynamics in the target shapes.

**Optimization Process.** We choose not to optimize the velocity field in the simulation for the ablation study. Since the initialized velocity $v_0^t$ is a unit vector, resulting in an unobvious observation, we manually scale the initialized velocity to a certain numerical number $\alpha$. In this case, we prepare the velocity field with the scaled velocity by parts, as $\boldsymbol{v^t} \leftarrow \alpha v_0^t$. On the other side, we set up the full experiment with the same velocity field and get both of the generated motions illustrated in Figure 7. It is noticed that without optimization, relative errors are accumulated for the motion, affecting the simulation to ill-posed states.

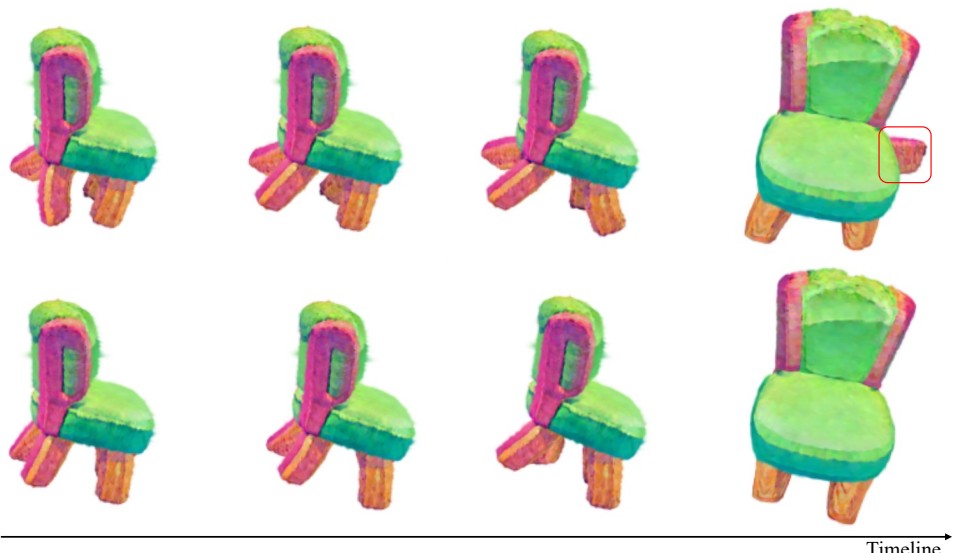

Timeline

Figure 7: Ablation study on optimization process. **Upper Row:** manually set up the initial velocity field. **Bottom Row:** with optimization to the initial velocity field.

## 5 CONCLUSION

This paper introduces Sync4D, a cutting-edge approach to 4D generation guided by casually captured video, which ensures exceptional motion realism and shape integrity. Our framework enhances general 3D generation by transferring motion with precise guidance from video sequences. Moreover, we incorporate physical simulations into the generation of 4D dynamics, optimizing the velocity field appropriately. Experimental results confirm the efficacy of Sync4D. This method not only facilitates intuitive control over 4D generation but also produces physically plausible dynamics, making it highly suitable for integration into various applications such as game engines and virtual reality environments.

**Limitations.** Although Sync4D is capable of generating diverse dynamics across various shapes and complex motions, it encounters difficulties when transferring continuous spinning motions. While Sync4D approximates revolute motions by segmenting the circular arc of rotation into multiple linear segments, spinning motions can be hard to deal with. The limitation arises due to challenges in accurately capturing and replicating such rapid, cyclical movements.

Our framework has a constraint regarding the alignment between the initial pose of the reference video and the generated 3D representation; significant deviations between the two can impact performance. This limitation stems from the model's focus on learning relative motion rather than replicating individual poses across frames. However, since our goal is to introduce motion controls to generated shapes, it is feasible to manage the initial pose during 3D generation or adjust the reference video's starting frame. Additionally, a pose alignment module could be incorporated in future work to address this limitation.

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

## A    APPENDIX

### A.1    MPM MATERIAL FIELD

Despite particle position $x$ and velocity $v$ being tracked in MPM simulation, particle material properties are also sufficiently needed for updating. Firstly, we go through how material property $F$, $C$, $\nu$, and $E$ can influence the deformation of the object. Our Gaussian model is viewed as a continuum mechanics model, who utilize a deformation map $\phi(\mathbb{X}, t)$ to record deformed space from base space $\mathbb{X}$. For numerical calculation, $F$ is introduced to store the deformation gradient of $\phi$, know as the Jacobian of the map:

$$F = \nabla_{\mathbb{X}} \phi(\mathbb{X}, t) \tag{17}$$

$F$ measures the local rotation and strain of the deformation and helps formulate the stress-strain relationship.

Another two physics parameters noted are Shear modulus $\mu$ and Lamé modulus $\lambda$, which are related to Young's modulus $E$ and Poisson's ratio $\nu$:

$$\mu = \frac{E}{2(1 + \nu)}, \quad \lambda = \frac{E\nu}{(1 + \nu)(1 - 2\nu)}. \tag{18}$$

These two parameters help formulate Kirchhoff stress $\tau$, which can be adapted to different elasticity and plasticity models. We utilize the fixed corotated elasticity model, whose Kirchhoff stress $\tau$ is defined as:

$$\tau = 2\mu(F^E - R)F^{E^T} + \lambda(J - 1)J, \tag{19}$$

where $F = F^E F^P$ is multiplicative decomposition on $F$, while $R = UV^T$ is a matrix from Singular Value Decomposition on $F$ as $F = U\Sigma V^T$. $J$ is the determinant of $F^E$.

In the process of MPM simulation, $F$, $C$, and $\tau$ are also updated in P2G, G2P process, which can be denoted as:

$$C_p^{t+1} = \frac{4}{r^2} \sum_i N(x_i - x_p^t)v_i^{t+1}, \tag{20}$$

$$F_p^{t+1} = (I + \Delta t C_p^{t+1})F_p^t, \tag{21}$$

$$\tau_p^{t+1} = \tau(F_p^{E,t+1}). \tag{22}$$

This is just one case application for MPM simulator and for more details, please refer to (Hu et al., 2018b; 2019; Jiang et al., 2017)

### A.2    USER STUDY RESULTS

We conduct the user study on three sets of experiments, which are from human to cross, from laptop to sea shell, and from human to monkey toy. Participants are asked to choose between renderings from

Table 1: Human study on Sync4D (Ours) over DMT generated video and DreamGaussian4D dynamics generation.

| Overall Visual Quality | human-to-cross | laptop-to-shell | human-to-monkey |
|---|---|---|---|
| Ours over DMT | 82.4% | 100% | 94.1% |
| Ours over DreamGaussian4D | 100% | 94.1% | 100% |
| *Motion similarity* | | | |
| Ours over DMT | 97.1% | 94.1% | 100% |
| Ours over DreamGaussian4D | 94.1% | 97.1% | 100% |
| *Shape consistency* | | | |
| Ours over DMT | 88.2% | 100% | 94.1% |
| Ours over DreamGaussian4D | 88.2% | 94.1% | 97.1% |

Sync4D and competitor's generation forcibly. The three evaluation metrics are *Overall visual quality*, *Motion similarity*, and *shape consistency*. We render our dynamics in a fixed view, comparing it to video motion transfer output and renderings of DreamGaussian4D. Table A.1 shows the remarkable advantage of Sync4D over other methods.

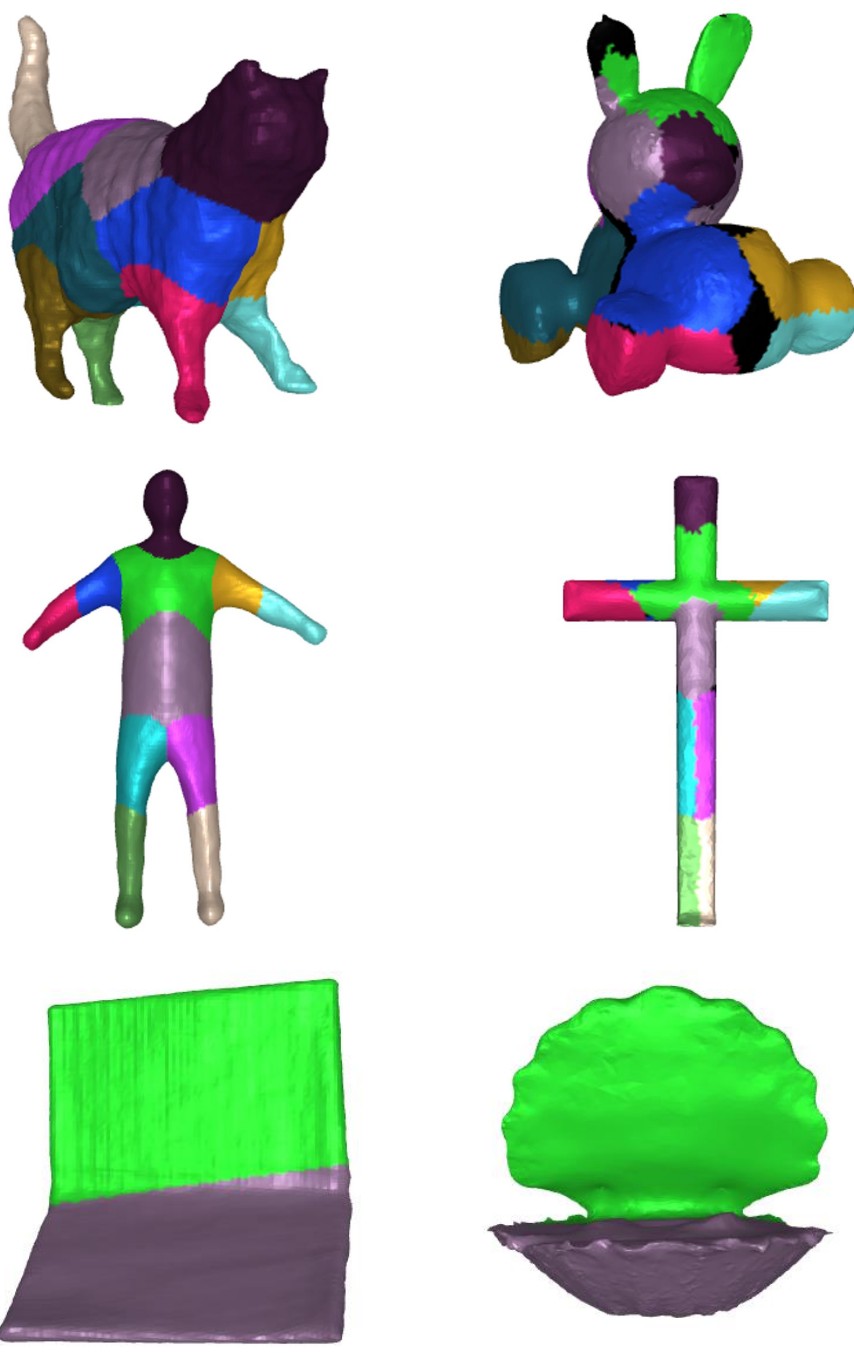

Figure 8: We showcase the articulated part matching between the reference and target shapes.

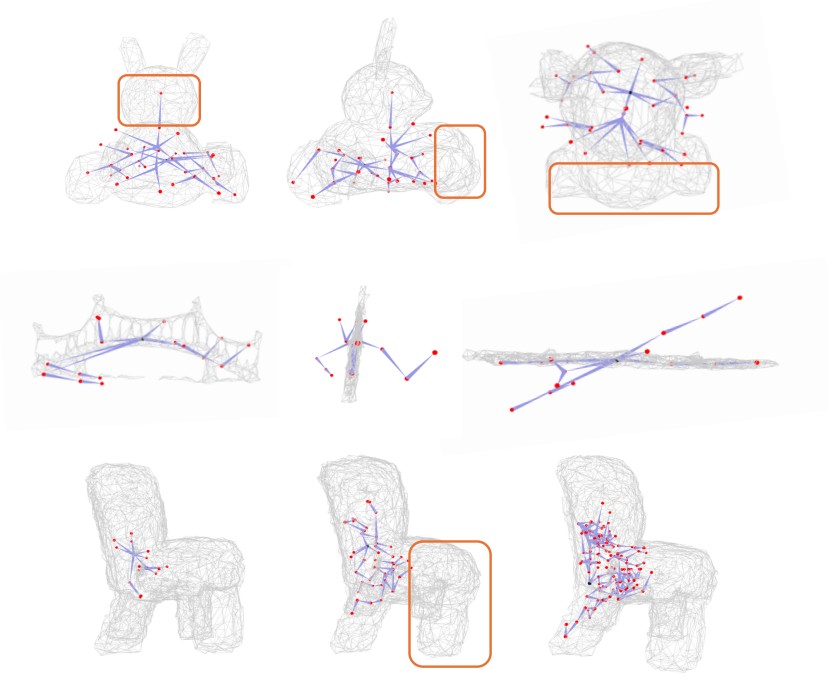

Figure 9: Auto rigging method RigNet fails on our generation.

### A.3 MATCHING DETAILS

**Matching Results.** In Figure 8, we present the results of articulated part matching between the reference and target shapes. Black color indicates the outliers that have been removed from the correspondence matching. As shown in row 2, for the human-cross pair, our method allows for reasonable matching even between pairs that are topologically different.

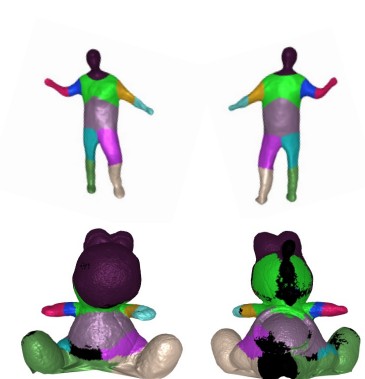

Figure 10: Correspondence between two different posed objects.

**Matching Sensitivity on Poses.** Our method robustly addresses pose mismatches through a sophisticated correspondence matching system, as illustrated in Figure 10. Our approach leverages both semantic and spatial features to establish correspondences. Semantic features are derived from advanced generative models such as DINO and Stable Diffusion, which capture rich semantic details. Additionally, we incorporate spatial features from CorrNet3D, a model specifically trained in a self-supervised manner to establish dense correspondences across shapes in varying poses. This dual-feature strategy ensures our correspondences are not only stable but also accurate, even across diverse and challenging poses.

**Failure Case.** Please refer to Figure 11 for failure case. In this human-tree case, not only does correspondence matching fail, but also motion is not guaranteed.

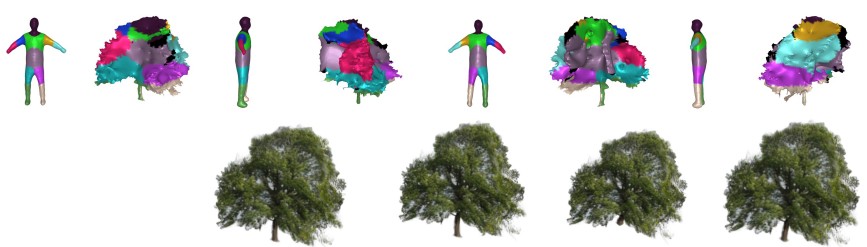

Figure 11: Failure case on human-tree matching and motion transfer.

## A.4 MORE QUALITATIVE RESULTS

We present additional motion transfer results involving shapes with different topologies and motions across various scenarios. Please refer to Figure 12

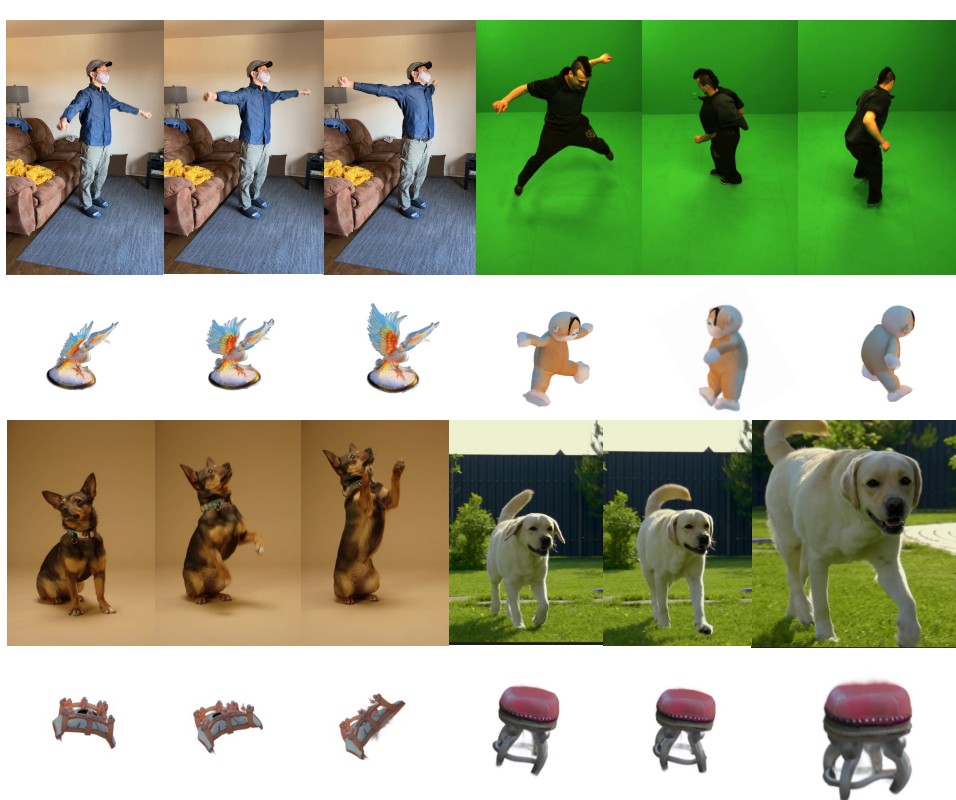

Figure 12: More qualitative results.

