# OpenReview forum: "Sync4D: Video Guided Controllable Dynamics for Physics-Based 4D Generation"
_ICLR.cc/2025/Conference — ICLR 2025 Conference Withdrawn Submission_

### Official Review · Reviewer_yaHa · 2024-10-31

**Soundness:** 2
**Presentation:** 1
**Contribution:** 2
**Rating:** 3
**Confidence:** 5

**Summary:**

This paper introduces an approach for dynamic 3D Gaussian generation from casual videos. Firstly, the shape and motion are extracted from the reference videos. The reconstructed objects are then segmented into motion-related parts based on skinning weights and mapped to the generated 3D Gaussians. Additional physical simulation and displacement loss are applied to drive the 3D Gaussians to maintain integrity and accuracy.

**Strengths:**

- This work is systematic and covers aspects from video-based geometry/motion reconstruction, 3D Gaussian optimization from diffusion prior, correspondence determination, to material point method-based motion transfer.

**Weaknesses:**

- The method seems to simply correspond the Gaussian points in the target shape to each part of the reference model, but it does not perform geometric analysis on the target shape or define a corresponding skeleton. The motion of the Gaussian points primarily relies on the movement of bones in the reference model and the solution of the material point method (MPM). This makes it difficult to ensure the reasonableness of motion under the target shape. For example, in the bottom left corner of Fig 5, the box shows significant distortion in the supplementary video (00:22s).
- The rendering quality of the generated dynamic 3D Gaussians is relatively poor. The advantage of Gaussians lies in rendering scenes with difficult surfaces, such as fuzzy geometry, while MPM excels at representing physical properties in motion, such as elasticity. However, the examples presented in this work do not demonstrate these strengths; all the dynamics are merely simple hard surface geometric deformations.
- Although Fig 9 mentions that many generated geometric rigs cannot be handled well, for some examples shown in this paper, such as the humanoid bear and giraffe, directly generating a 3D mesh, coupled with geometric analysis to extract the skeleton and skinning weights and then applying texture mapping, could yield better results.

**Questions:**

Please refer to the weakness.

---

### Official Review · Reviewer_2jKq · 2024-11-03

**Soundness:** 3
**Presentation:** 3
**Contribution:** 2
**Rating:** 5
**Confidence:** 4

**Summary:**

The author proposes a controllable 4D generation pipeline by transferring a source video motion to a target 3D object. To achieve this goal, the author proposes a a part mapping pipeline to match shape between source and target object,  and further integrate with MPM-guided physical simulation to synthesize the deformation. Experimental results show good motion transfer on various types of objects.

**Strengths:**

**Interesting application**

The application is quite interesting and the generated results are good. Free-form motion transfer for 4D generation is useful in content creation community.


**Motion transfer with physics integrated**

It is good to integrate MPM into motion transfer, with such we could generate deformations that typical articulated models can't achieve (like parametric human and animal models). The proposed velocity field optimization achieved good visual results.


**Applicable to different objects**

In the video the author shows good generalization to different types of objects.

**Weaknesses:**

**Shape correspondence**

The matching only relies on shape correspondence. I wonder how could the matched parts guarantee to have good motion similarities with only rely on static features. Some more results on articulated objects and different geometry should be included to demonstrate the effectiveness, it will be also good to report some quantitative metric like matching success rate.




**MPM VS point-wise deformation**

The author mentions point-wide deformation would cause temporal inconsistency, however the author doesn't do such ablations to support the claim. For MPM, similar to what Zhang et al 2024 did, the author doesn't optimize the Young’s modulus and other physical parameters, so it is hard to say the later MPM-based method actually follows the real physics motion and thus weaken the built deformation. I am not fully convinced why MPM would work generally better than point or skeleton based approaches. Besides, MPM would be very hard to maintain rigidity for rigid objects and will also not work well for articulated objects and human with clear skeleton geometry.



**Missing quantitative results**

As the author claim to do motion retargeting, the author should report some common metrics in motion transfer for ablations (like vertex-to-vertex distance and others) of different design choices and compare  with different type of objects (rigid, articulated, soft etc).

**Questions:**

**Video motion**

It seems the author uses BANMo to extract source object motions and geometry. However, BANMO might not suitable for all casually captured videos. I wonder what is the bottleneck in such pipeline and could the extracted motion well reflect the real object motions. For a video with both camera motion and object motion, will the transfer be success?

---

### Official Review · Reviewer_aQpe · 2024-11-04

**Soundness:** 3
**Presentation:** 3
**Contribution:** 2
**Rating:** 5
**Confidence:** 4

**Summary:**

The paper proposes a novel method to transfer motion from casually captured monocular videos to generated 3D models implicit representations and bones with skinning weights. A mapping is computed using deep features in 2D and 3D between the casually captured 3D object and the generated 3D object. Temporal consistency is achieved using a physically based MPM simulation framework used commonly in computer graphics.

**Strengths:**

The paper is written well and method's originality is utilizing the MPM framework to retarget motion which makes this method temporally consistent and physically based. The proposed method is evaluated on real-world reference videos to drive synthetic objects and is able to transfer motion between very dissimilar topologies while maintaining temporal consistency with the use of a triplane velocity field to account for accumulated errors.

**Weaknesses:**

- Various key decisions of the technique are not thoroughly evaluated. For example, with/without the Triplane velocity field, effects of various loss components, etc.
- The method is qualitatively evaluated on a limited dataset of reference videos and generated set of 3D objects. Expanding this evaluation to some publicly available datasets would improve quality of evaluation.
- There's no information around approximate training and inference speed for the method.

**Questions:**

- Is it possible to infer material properties like elasticity from the reference video?
- Is it possible to retarget to real-world objects in addition to synthetic objects?
- Could you clarify the training methodology around how the different phases of the pipeline are trained and used to produce the final output?

---

### Official Review · Reviewer_3WZ1 · 2024-11-04

**Soundness:** 3
**Presentation:** 2
**Contribution:** 3
**Rating:** 5
**Confidence:** 4

**Summary:**

Sync4D generates controllable 4D dynamics in 3D Gaussian representations, guided by casually captured reference videos. This approach uses blend-skinning-based non-parametric shape reconstruction to extract shape and motion from reference videos. It transfers motion to generated 3D Gaussian objects across categories by establishing shape correspondences and mapping motion-related parts. To tackle shape and motion inconsistencies, it incorporates physical simulation optimized with a displacement loss, ensuring reliable and realistic dynamics. Sync4D supports diverse inputs—humans, quadrupeds, and articulated objects—and claims superior performance over diffusion-based video generation methods.

**Strengths:**

- Approach: This method uniquely transfers motion from casual videos to 3D objects, bridging 2D video inputs with 4D dynamic generation. Using the Material Point Method (MPM) for simulation, it heightens realism and physical accuracy. The framework supports diverse object categories and motions—humans, animals, and articulated objects—showcasing robustness. A displacement loss optimizes the velocity field, reducing cumulative errors and improving temporal and shape consistency.
- Extensive Experimentation: Experiments cover qualitative results, comparisons, ablations, and user studies, validating the approach's effectiveness.

**Weaknesses:**

- Related Works: The paper fails to reference several works in the field, such as Transfer4D, SC4D (Sparse-Controlled Video-to-4D Generation), and MagicPose4D. These works are directly related to the topic and have introduced techniques in motion transfer and 4D generation that are relevant to the proposed method. The lack of discussion about these works limits the contextualization of the contributions and may overlook existing solutions to some of the challenges addressed.
- Clarity: The paper's structure could be improved for better readability. Some sections, particularly the methodology, are dense and may be challenging for readers unfamiliar with the background concepts.
- Lack of Quantitative Eval: While qualitative results and user studies are provided, the paper lacks quantitative metrics (e.g., numerical comparisons, statistical significance) to objectively assess performance against baseline methods.
- Limited Discussion on Limitations: The limitations section briefly mentions challenges with spinning motions and initial pose alignment but lacks an in-depth analysis or potential solutions.
- Comparative Analysis: The comparisons with existing methods seem somewhat limited. For instance, the failure cases of other methods are highlighted, but it would be beneficial to include more successful cases for a balanced evaluation.

**Questions:**

Q1: Can you provide quantitative metrics to evaluate the performance of your method against baseline models? This would strengthen the validation of your approach.

Q2: Could you elaborate on the implementation details of the triplane representation and the optimization process for the velocity field?

Q3: Beyond the brief mention, could you provide a deeper analysis of the limitations of your method, particularly regarding spinning motions and initial pose alignment? Are there potential solutions or future work directions to address these issues?

**Details Of Ethics Concerns:**

N/A.

---

### Note · Authors · 2024-11-12

I have read and agree with the venue's withdrawal policy on behalf of myself and my co-authors.